



# A small-scale and autonomous testbed for three-line delta kites applied to airborne wind energy

Francisco DeLosRíos-Navarrete[1,2], Jorge González-García[2], Iván Castro-Fernández[3], and Gonzalo Sánchez-Arriaga[2]

[1]CT Ingenieros A.A.I. S.L. Avenida Leonardo Da Vinci 22, 28050 Getafe (Madrid), Spain
[2]Department of Aerospace Engineering, Universidad Carlos III de Madrid, Avenida de la Universidad 30, 28911 Leganés (Madrid), Spain
[3]Department of Space Programmes, Instituto Nacional de Técnica Aeroespacial, Carretera de Ajalvir Km. 4, 28850 Torrejón de Ardoz (Madrid), Spain

**Correspondence:** Francisco DeLosRíos-Navarrete (francisco.delosrios@ctengineeringgroup.com)

**Abstract.**

A mechanical control system and the guidance and control modules of a small-scale and autonomous testbed for three-line kites applied to airborne wind energy are presented. It extends the capabilities of a previous developed infrastructure by (i) changing the actuation system to add a third tether to control the kite pitch angle, (ii) adding running line tensiometers to measure the three tether tensions while allowing tether reel-in and reel-out, and on-board load cells to measure the bridle tensions, (iii) a real-time control system to operate the kite autonomously in figure-eight trajectories. A controller based on a hybrid guidance scheme for figure-eight flight paths, which combines attractor points for the straight segments and a continuous heading angle tracking for the turns, was implemented and validated in an experimental campaign. Two flights of the campaign were used to illustrate the performance of the controller and its capability to adjust the lateral amplitude, elevation and radius of the turns by varying a few parameters of the guidance module. The proposed control scheme was proven effective in achieving satisfactory and repeatable figure-eight paths. The experimental data collected during the autonomous flight was used to investigate the dynamics and control of the kite and the tethers. A correlation between the heading and roll angles of the kite was identified and modeled with a simple analytical law with empirical coefficients. Similarly to previous works on airborne wind energy, a linear relation between the derivative of the course angle and the steering input was found. The analysis of the on-ground tensiometers and the on-board load cells revealed a variable time delay up to 0.2 s between both measurements. The work shows that the testbed and its instruments are suitable to investigate the effect of tether sagging and to develop and test controllers for airborne wind energy systems.

## 1 Introduction

The availability of the wind resource is a determining factor for the economic viability of wind energy power sources (Coca-Tagarro, 2023). When compared to traditional wind turbines, Airborne Wind Energy (AWE) systems could increase the availability of the 5th percentile wind power density by a factor of 2 over most of Europe (Bechtle et al., 2019). Despite its potential,





AWE systems face a particular set of challenges due to their nature as autonomous flying devices with operational areas expanding hundreds of meters from the ground station. Safety and reliability are identified as some of the main concerns affecting the social acceptance of the technology (Schmidt et al., 2022). Consequently, the aerodynamic characterization of kites and the experimental validation of control algorithms are key for the deployment of the technology, and most of AWE companies are currently focused on the long-term and repeatable operation of the machines (Kitemill, 2023; Kitekraft, 2023). The recent measurement of the SkySails PN-14 System power curve based upon the standard IEC 61400-12-1 (Bartsch et al., 2024) is an important milestone for AWE.

A large number of works presented experimental results of AWE systems, or of their fundamental components, in the past. A number of approaches to experimental testing can be found in the literature. In-lab testing facilities have been successfully used for gathering data from scaled models, often based on water channel setups for improved dynamic similarity (Cobb et al., 2018) or specialized in takeoff and landing (Azaki et al., 2023). Tow-tests, in which a kite is attached to a moving vehicle to emulate the wind flow, have also been extensively used (Wood et al., 2017; Hummel et al., 2019), with some examples aimed at kite performance measurements predating most works on AWE (Alexander and Stevenson, 2001). Nonetheless, a great number of research groups have focused on the development of ground-fixed prototypes for field tests in a plethora of configurations, often in close collaboration with AWE companies. Some experimental setups have been developed for conducting research on specific topics, like kite aerodynamic characterization (Borobia-Moreno et al., 2021) or autonomous takeoff and landing (Fagiano et al., 2022), while others are multipurpose rigs used for aerodynamic characterization (Oehler and Schmehl, 2019), and guidance and control research (Ahrens et al., 2013), among other topics.

A pillar of many of the previously mentioned experimental setups is to study and validate control strategies for the autonomous flight of AWE systems. One relatively simple yet extensively tested guidance strategy to achieve figure-eight crosswind path is the use of attractor points. As opposed to continuous (Diwale et al., 2017) or discrete (Wood et al., 2015) parametrizations of the desired trajectory, in which the controller aims to steer the kite in a strictly defined path, in this approach the kite is guided towards a reduced number of waypoints, which are sequentially switched to follow a lemniscate path. This strategy has been validated in several experimental setups using leading edge inflatable kites, both ground-actuated (Fagiano et al., 2014) and fly-actuated (Fechner and Schmehl, 2016). However, to the best of the author's knowledge, no examples can be found on the literature on the experimental use of this strategy on three-line rigid-framed delta kites. Rigid wings are particularly interesting for AWE applications due to some aerodynamic advantages as compared to leading edge inflatable or foils kites (Cherubini et al., 2015). Rigid wings, however, present some drawbacks like their inferior robustness to impacts during takeoff and landing and pose challenges to achieve robust control due to their highly dynamic response and the presence of non-stationary aerodynamic phenomena (Castro-Fernández et al., 2024). The AWE machine of EnerKite GmBH (Bormann et al., 2013; Candade et al., 2020) is a practical example of the potential application of the control strategy for three-line delta wings studied in this work.

This study on the autonomous control of a rigid-framed delta kite was conducted by first improving the capabilities of the automatic ground station of Universidad Carlos III de Madrid (Castro-Fernández et al., 2023) and then performing several test campaigns. As explained in Sec. 2, a new configuration of the control system was proposed and a second linear actuator was



added to allow for pitching control of the kite. The amount and quality of the scientific data that can be collected by the testbed were also improved by adding more sensors. For instance, on-board load cells to measure the tether tensions directly applied to the kite were included. A hybrid guidance strategy is proposed in Sec. 3. Based on a controller, it uses attractor points for

the straight segments of the figure-eight path and a continuous angle tracking for the curved segments. The results of two flight tests of an experimental campaign, including a 5-minute autonomous flight, are presented and discussed in Sec. 4. Finally, the conclusions are presented in Sec. 5.

## 2 Small-scale control system

### 2.1 System architecture

As shown in Fig. 1a, the proposed system is composed of a ground control unit (GCU) and a rigid-framed delta (RFD) kite equipped with on-board sensor hardware. The RFD kite's design is based on a HQ Kites™ model Fazer XXL and is connected to the GCU by a set of three Dyneema® tethers, routed through a system of pulleys to a common winch mechanism (see Fig. 1b). The drum is fitted with a 3D-printed grooved sleeve to passively guide the tethers during winding. A set of intermediate linear actuators allows for the independent control of the lengths of the left and right tethers (hereafter called the control

lines). Both the control lines and the central tether pass through independent running line tensiometers, equipped with load cells to measure tether tensions even if the winch mechanism is actuated. All elements are fixed to a common aluminum base plate through an assortment of aluminum and steel supports, some of which are equipped with quick-connect mechanisms to facilitate transportation and maintenance. The specifications of the linear actuators, servomotors and the winch motor are as described in Castro-Fernández et al. (2023). The powertrain has been upgraded, incorporating a gearbox with a reduction ratio

of 20:1 and a winch radius of 49 mm.

Figure 2 shows a block diagram of the elements used in the GCU and the RFD kite. Green, blue, orange and gray colors were used to denote sensors, active signal-processing components, actuators, and human-machine interfaces, respectively. Dashed and dashed-dotted lines were used to denote wired and wireless connections between elements, while mechanical links are shown with solid lines. The next subsections explain in detail each of these building blocks.

### 80 2.2 Mechanical Control System

Control over the steering and pitch angle of the RFD kite is provided by the proposed control system through the coordinated movement of the linear actuators. Some frames of reference and geometric characteristics should be introduced to understand the principles of actuation of the system. As shown in panel (b) of Fig. 1, we call $C_L$ and $C_R$ to the points where the pulleys of the left and right linear actuators are placed. At points $O_L$ and $O_R$ the pulleys of the left and right actuators reach their

minimal distances to the winch. A frame of reference $S_G$ is introduced, with origin $O_G$ located at the intersection between the central tether and the virtual line that passes through points $O_L$ and $O_R$, $x_G$ pointing downwind and $z_G$ pointing upwards. The displacement of the linear actuators is given by the coordinates $x_L$ and $x_R$ of points $C_L$ and $C_R$, respectively. The origin for





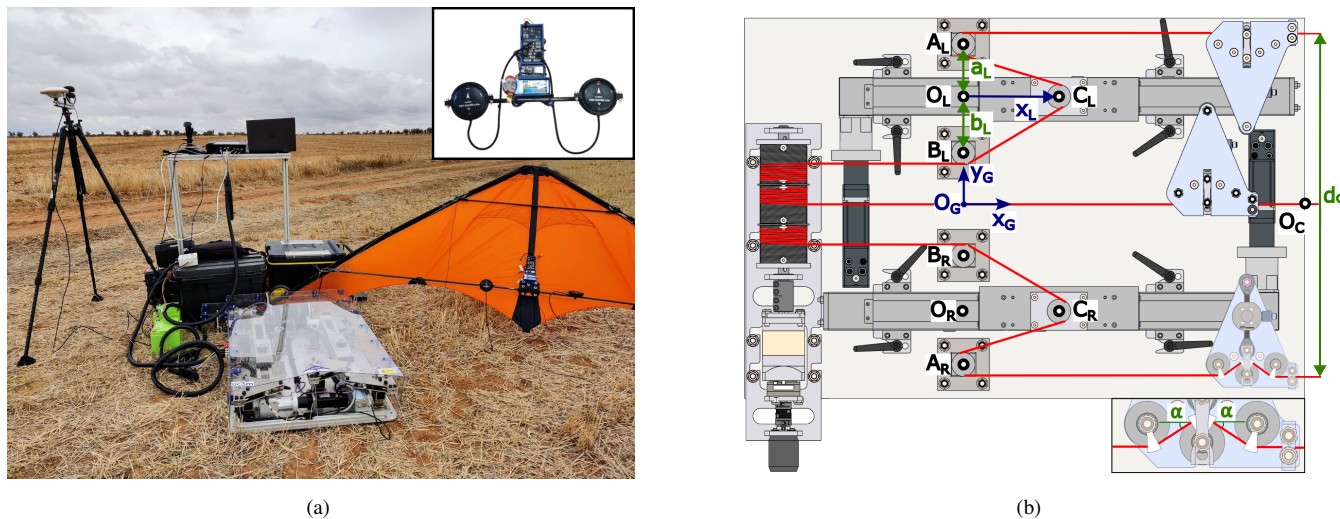

(a)                                               (b)

**Figure 1.** (a) The GCU and the RFD kite. The inset shows a detail of the on-board electronics. (b) Diagram of the GCU, highlighting the tether's path, the ground reference frame (denoted by subscript $G$) and several geometric points. One tensiometer cover was removed to show its internal mechanism. The inset shows the definition of the tether angle $\alpha$.

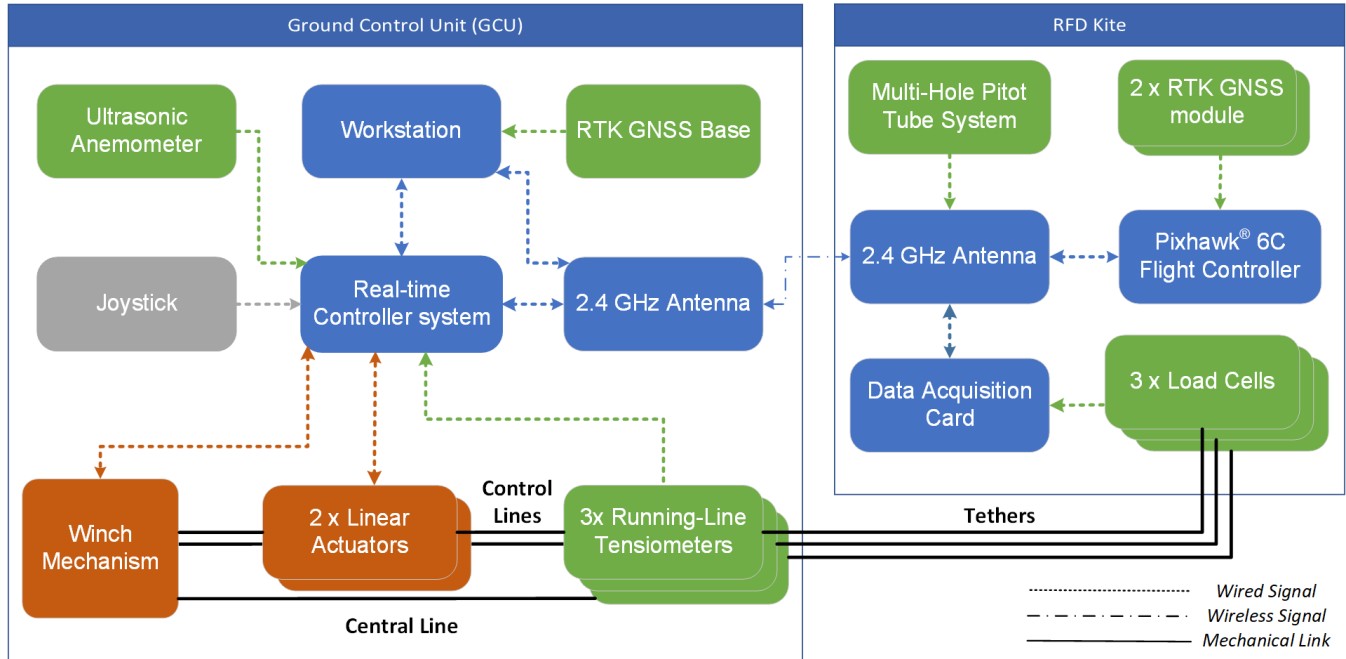

**Figure 2.** Sensors (green), active signal-processing components (blue), actuators (orange), and human-machine interfaces (gray) of the GCU (left) and the kite (right).





the controller's spherical frame of reference, defined in Sec. 3, is given by $O_C$, which is placed at the intersection between the $x_G$-axis and the edge of the GCU. Two important design parameters that define the geometry of the control system are

$$a_j = \|\overline{O_j A_j}\|, \quad b_j = \|\overline{O_j B_j}\|, \quad j \in \{L, R\}, \tag{1}$$

that are the distances between points $O_L$ (or $O_R$) and the points $A_L$ and $B_L$ (or $A_R$ and $B_R$) where two auxiliary pulleys are located. The control is able to steer the kite because the displacement of the linear actuators $x_j$ produces a variation of tether length $\ell_j = |A_j C_j| + |C_j B_j|$. The variation of the distance with respect to the reference state without actuator displacement is

$$\Delta\ell_j(x_j) = \ell_j(x_j) - \ell_j(0) = \sqrt{x_j^2 + a_j^2} + \sqrt{x_j^2 + b_j^2} - a_j - b_j, \quad j \in \{L, R\}. \tag{2}$$

For later use, it is convenient to write the actuator displacements $x_j$ as a function of the tether length variation $\Delta\ell_j$. From Eq. (2) one finds

$$x_j(\Delta\ell_j) = \frac{\sqrt{\Delta\ell_j(2a_j + \Delta\ell_j)(2b_j + \Delta\ell_j)(2a_j + 2b_j + \Delta\ell_j)}}{2(\Delta\ell_j + a_j + b_j)}, \quad j \in \{L, R\}. \tag{3}$$

Both actuators share a common neutral position $x_0$, defined as the place of the pulley where the carriages rest when no steering input is commanded. The neutral position $x_0$ is initially defined at the middle of the actuator's physical range and can be dynamically adjusted during the flight to control the pitch angle of the RFD kite. When $x_0$ is changed, the distance between the kite's control anchoring points and the ground station is modified, while the distance to the central anchor point remains unchanged, consequently inducing a pitch on the kite.

Kite steering is achieved through the control input $\Delta L_u$, defined as the difference in length between the control tethers outside the GCU ($L_R - L_L$), which coincides with the tether length difference inside the GCU. Therefore, one has

$$\Delta L_u = L_R - L_L = \ell_L - \ell_R. \tag{4}$$

The proposed scheme uses variable $x_0$ to control the pitch of the kite and $\Delta L_u$ to steer it. Such an approach is convenient because each control input is in charge of commanding a different degree of freedom of the kite. For given values of $x_0$ and $\Delta L_u$, one first finds the quantities

$$\Delta\ell_L(x_0, \Delta L_u) \equiv \ell_L(x_0) + \frac{\Delta L_u}{2} - \ell_L(0), \quad \Delta\ell_R(x_0, \Delta L_u) \equiv \ell_R(x_0) - \frac{\Delta L_u}{2} - \ell_R(0), \tag{5}$$

and their substitution in Eq. (3) provides the displacements of the actuators.

### 2.3 On-ground tensiometers and on-board load cells

The GCU has three running line tensiometers. These devices provide an indirect measurement of the tether tension while simultaneously allowing for reel-in and reel-out. As shown in Fig. 1b, the tether is routed through a set of three pulleys inside the tensiometer, two of which are fixed while the central one is connected to the load cell with a rigid link. The contact surface between the link and the chassis of the tensiometer restricts the movement of the pulley to the direction perpendicular to the tether. The traction forces $F_{LC_j}$ measured by the load cells are

$$F_{LC_j} = 2\, k_j\, T_j\, \sin\alpha, \quad j \in \{L, C, R\}, \tag{6}$$



| Parameter | Value |
|-----------|-------|
| $a_L, a_R$ | $0.110\ m$ |
| $b_L, b_R$ | $0.118\ m$ |
| $d_c$ | $0.722\ m$ |
| $\alpha$ | $30°$ |
| $k_L$ | $0.847$ |
| $k_C$ | $0.781$ |
| $k_R$ | $0.833$ |

**Table 1.** Characteristic parameters of the GCU

where $T_j$ is the tether tension and $\alpha$ is the entry and exit angle of the tether shown in the inset of Fig. 1b. The empirical and dimensionless factor $k$ takes into account the internal friction of the tensiometer that appears in the pulleys and between the mobile pulley and the structure. Factor $k$ was calibrated for each tensiometer with a static load test in which a tether was fixed to an independent load cell on one side and to a set of known loads on the other. Table 1 shows the measured values of $k$ for the three tensiometers and the geometrical parameters of the GCU appearing in Eq. (2).

The three tensiometers located at the GCU measure the tether tension on the ground. However, since the tether is subjected to acceleration and other forces like gravity and the aerodynamic force, the tether tensions at the kite are different. For this reason, three small load cells were added on-board the kite. Unlike the tensiometers of the GCU, which needs to be compatible with tether reel-in/out, the on-board load cells were directly located between the tether tip and the bridles. Having simultaneous knowledge of the tether tensions on the ground and at the kite open the possibility of investigating some interesting topics like for instance the impact of aerodynamic load on the tether dynamics and also provide useful data to validate tether models in AWE simulators.

## 2.4 Electronic system architecture

As shown in Fig. 2, the RFD kite is equipped with a Pixhawk® 6C flight controller, used for logging the kite kinematic state. The controller fuses the measurements from its embedded IMU, magnetometer and barometers with the reading from the RTK GNSS modules in a built-in Kalman filter. A custom-made data acquisition board samples the signals coming from the on-board load cells at regular intervals. A multi-hole pitot tube system, as the one used in Ref. (Borobia-Moreno et al., 2021), can also be integrated on the platform to gather data for aerodynamic analysis, although it was not incorporated for this flight campaign as it was not the focus for this work. All on-board electronics are powered by a 7.4 V LiPo battery and a 5 V DC/DC converter. A communication link with the GCU is made with a pair of 2.4 GHz antennas.

The GCU's main control board is based on the Texas Instruments™ F28379D real-time microcontroller. The built-in CAN Bus transceiver is used to communicate with both linear actuator servomotors and the winch mechanism motor controller. Measurements from the ultrasonic wind station and the running line tensiometers are also logged in real time. A joystick is





present to provide manual control during takeoff and landing maneuvers, adjust the controller parameters and set open or close loop control as desired by the operator. A workstation is used to log all data gathered by the control board and to broadcast RTK data from the GNSS base station. The power for the GCU is supplied by two 12 V lead-acid batteries connected in series.

## 3 Controller design

The proposed controller uses three angles, which are represented in Fig. 3. Two of them, $\lambda$ and $\delta$, are the elevation and azimuth of the position vector $\boldsymbol{r}$, which has origin at point $O_C$ of the GCU and its tip at the center of mass $O_K$ of the kite. It reads

$$\overline{O_C O_K} = r\left(\cos\lambda\cos\delta\boldsymbol{i}_G + \cos\lambda\sin\delta\boldsymbol{j}_G + \sin\lambda\boldsymbol{k}_G\right) \tag{7}$$

where $\boldsymbol{i}_G$, $\boldsymbol{j}_G$ and $\boldsymbol{k}_G$ are the unit vectors of the $S_G$ frame and $r$ is the distance between point $O_C$ and the center of mass of the kite. Therefore, $\lambda$ is the elevation angle with respect to ground, and $\delta$ measures the lateral displacement of $O_K$ with respect to

the $x_G - z_G$ plane. The third angle used by the controller is the heading angle $\psi$, which is defined as

$$\tan\psi \equiv \frac{\boldsymbol{i}_K \cdot \boldsymbol{u}_\delta}{\boldsymbol{i}_K \cdot \boldsymbol{u}_\lambda} = \frac{\boldsymbol{i}_K \cdot (\sin\delta\boldsymbol{i}_G - \cos\delta\boldsymbol{j}_G)}{\boldsymbol{i}_K \cdot (-\sin\lambda\cos\delta\boldsymbol{i}_G - \sin\lambda\sin\delta\boldsymbol{j}_G + \cos\lambda\boldsymbol{k}_G)}, \tag{8}$$

with $\boldsymbol{i}_K$ the unit vector along the direction defined by the spine of the RFD kite, and vectors $\boldsymbol{u}_\delta$ and $\boldsymbol{u}_\lambda$ defined by the last equality in Eq. (8). The heading angle is measured on the tangent plane defined by the meridian ($\boldsymbol{u}_\lambda$) and parallel ($\boldsymbol{u}_\delta$) unit vectors using as reference the vector $\boldsymbol{i}_K$ defined by the kite's spine. Since $\psi$ is the angle between a meridian and the $x_G$-axis of the kite, $\psi$ vanishes when the kite is pointing towards the North pole of the sphere in Fig. 3.

As shown in Fig. 3b, the figure-eight path is divided into two straight segments and two turning sections. Each straight segment is defined by a reference attractor point ($R_\pm$) and a transition condition based on azimuth ($\delta_{L_\pm}$). Each turning maneuver is defined by a reference center point $C_\pm$ and a transition condition based on heading angle $\psi_{L_\pm}$.

Figure 4 shows the architecture of the controller. For clarity, we separate it into five main parts connected sequentially: (i) the guidance module that receives the angular coordinates of the center of mass ($\delta, \lambda$) and the heading angle ($\psi$) and finds a

heading angle setpoint ($\psi_{sp}$), (ii) a PID controller that produces the tether length difference setpoint $\Delta L_{u_{sp}}$ from the output of the guidance module, (iii) a transformation block that finds the angular position setpoint of the motors ($\theta_{L_{sp}}$ and $\theta_{R_{sp}}$), (iv) a built-in cascade controller for each motor that computes the required current ($i_L$ and $i_R$) to set the actuators at the angular positions $\theta_L$ and $\theta_R$, and (v) the gain blocks that convert the motion of each motor into variations of tether distance.

Two main assumptions have been made for the design of the controller. Firstly, the radial and tangential motion of the kite on the wind sphere are considered to be decoupled, which is common practice in the literature (Rapp et al., 2019) and allows to study the control of steering maneuvers independently of the actuation on the winch mechanism and the radial coordinate $r$. Secondly, the heading angle is assumed to be approximately equal to the actual course angle of the RFD kite during crosswind conditions. The course angle is computed like the heading angle in Eq. 8 but replacing $\boldsymbol{i}_K$ by the absolute velocity vector of the

kite. This assumption allows to use the same control variable both for figure-eight maneuvers and for the hovering safe mode, in which the kite is positioned on top of the wind sphere and its absolute velocity is close to zero.



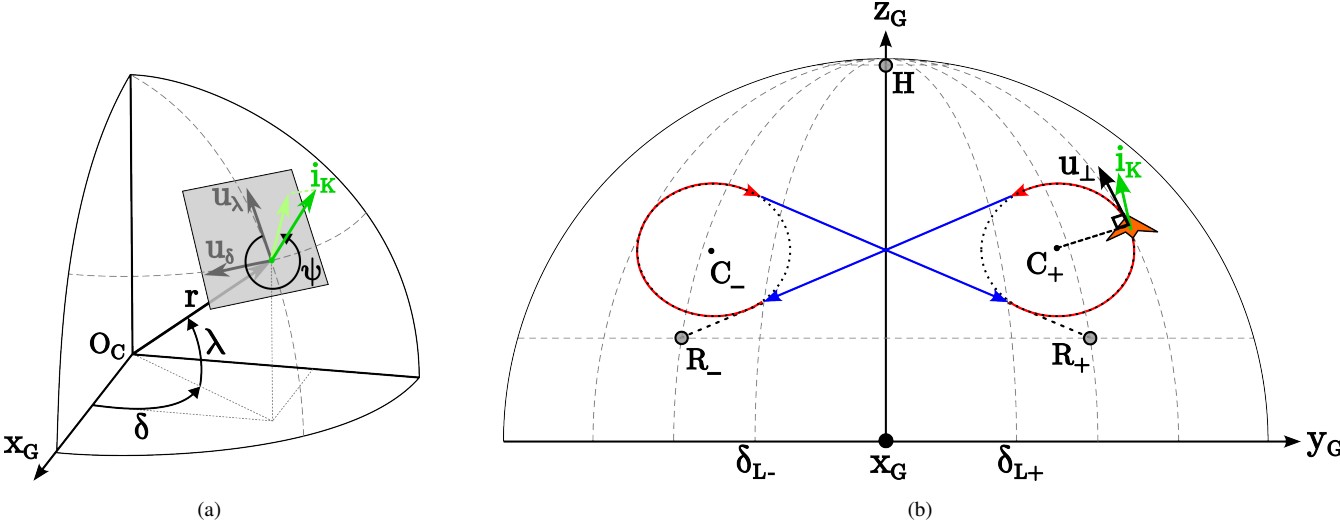

(a)                                                    (b)

**Figure 3.** (a) Elevation ($\lambda$), azimuth ($\delta$) and heading ($\psi$) angles of the RFD kite. (b) The reference figure-eight trajectory of the kite, which is divided into two downwards straight segments (blue) targeting a reference point ($R_\pm$), and two upwards turning segments (red) circling a reference center ($C_\pm$). The reference point $H$ for the hovering mode, the transition limits $\delta_{L\pm}$ and the auxiliary vector $\boldsymbol{u}_\perp$ perpendicular to the turning radius are also shown.

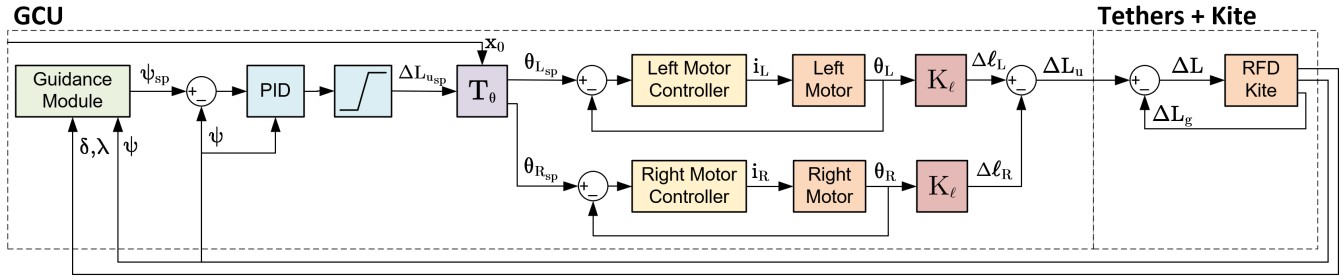

**Figure 4.** Block diagram of the control system, including a Guidance Module (green), a $\Delta L_u$ PID setpoint controller (blue), a transformation block (purple), two motor cascade controllers (yellow), two gain blocks (red) and the controlled plants (orange).

A finite-state machine approach was used to define the behavior of the Guidance Module. Accordingly, angle $\psi_{sp}$ in Fig. 4 takes different values depending on the mode of operation (figure-eight or hovering) and the specific segment (straight or turn) in the figure-eight trajectory. Both for the straight segment and for the hovering mode the heading angle setpoint $\psi_{sp,s}$ is

defined as the angle that the kite should take to be pointed directly towards an attractor point ($R_\pm$ and H). Conversely, during the turning segments, $\psi_{sp,t}$ is defined such as the kite is oriented perpendicular to the vector $\overline{C_\pm O_K}$ and pointing accordingly the up-turning flight trajectory (see Fig. 3b). The initial values of these reference points and thresholds for each flight are described in Table 2. The finite-state machine transition from the straight path to the turn is based on the difference $\delta - \delta_{L\pm}$, and, from the turn to the straight path, by monitoring $\psi - \psi_{L\pm}$.





The value of $\psi_{sp}$ is calculated according to the great-circle navigation formulas (Fechner and Schmehl, 2016)

$$\tan\psi_{sp,s} = \frac{-\sin\left(\delta_{R_\pm} - \delta_{O_K}\right)\cos\lambda_{R_\pm}}{\cos\lambda_{O_K}\sin\lambda_{R_\pm} - \sin\lambda_{O_K}\cos\lambda_{R_\pm}\cos\left(\delta_{R_\pm} - \delta_{O_K}\right)},\tag{9}$$

$$\tan\psi_{sp,t} = \frac{-\sin\left(\delta_{O_K} - \delta_{C_\pm}\right)\cos\lambda_{C_\pm}}{-\cos\lambda_{O_K}\sin\lambda_{C_\pm} + \sin\lambda_{O_K}\cos\lambda_{C_\pm}\cos\left(\delta_{O_K} - \delta_{C_\pm}\right)} + \frac{\pi}{2}sgn\left(\delta_{O_K}\right),\tag{10}$$

where $\delta_j$ and $\lambda_j$ with $j = O_K$, $R_\pm$ and $C_\pm$ are the azimuth and elevation of the three points (see Fig. 3). Angle $\psi_{sp,t}$ does not involve the turning radius because the proposed approach does not impose a predefined path. It rather aims to achieve a smooth circular trajectory independently of the kite's position at the end of the straight segment.

A digital PID controller is used to compute $\Delta L_{u_{sp}}$. The transfer function in the Z-domain of the PID is

$$U_{PID}(z) = \left(K_p + K_i\frac{T}{1-z^{-1}}\right)E_\psi(z) - \left(K_d\frac{1-z^{-1}}{T}\frac{1-d}{1-dz^{-1}}\right)\psi(z),\tag{11}$$

where $K_p$, $K_i$ and $K_d$ are the proportional, integral and derivative gains respectively, $T$ is the sample period of the controller, $E_\psi(z) = \psi_{sp} - \psi$ is the sampled heading angle error and $d$ is the decay value constant of the low-pass single-pole infinite impulse response filter used for the derivative input term. A derivative-on-measurement scheme, which computes the derivative term from the measured process variable instead of the error, is used to avoid derivative kick effects at the transition between states. $T$ is imposed by the sample frequency of the telemetry from the Flight Controller, while the rest of parameters were first tuned prior to the flight based on a simulation work performed with the LAKSA software (Sánchez-Arriaga et al., 2021)(de-losRíos Navarrete et al., 2023). Nevertheless, all the parameters can be adjusted during flight and some of them were modified to improve the performance of the controller. The baseline parameters used for the flights described in Sec. 4 are given in Table 3. As shown in Fig. 4, the output of the controller is limited by a saturation function, which ensures $\Delta L_{u_{sp}}$ is always within the actuator's achievable range.

The transformation block $T_\theta$ converts $\Delta L_{u_{sp}}$ into angular setpoints for each servomotor. By combining Eqs. (3) and (5) with a constant conversion factor $k_{x\theta}$ to account for the mechanical relationship between the angular movement of the motors and the lineal displacement of its actuator's carriages we yield the following transfer function

$$T_\theta\left(\Delta L_{u_{sp}},\ x_0\right) = k_{x\theta}\begin{bmatrix} x\left(\Delta\ell_L(x_0, \Delta L_{u_{sp}})\right) \\ x\left(\Delta\ell_R(x_0, \Delta L_{u_{sp}})\right) \end{bmatrix}.\tag{12}$$

The built-in cascade controllers of the motors used to reach the desired $\theta_{sp}$ are based on a PID position controller whose output is fed into a PI current regulator. Both controllers apply anti-windup methods. A feedforward of the angular speed and acceleration setpoints is used to compensate velocity-proportional friction and the inertia, respectively. According to the manufacturer's documentation (Maxon, 2021), the output of the position controller including the feedforward terms are modeled by the transfer function

$$C_P(s) = \left(K_{p_P} + \frac{K_{i_P}}{s} + \frac{K_{d_P}s}{1 + \frac{K_{d_P}}{10K_{p_P}}s}\right)E_\theta(s) + FF_\omega\ R_\omega(s) + FF_\alpha\ R_\alpha(s),\tag{13}$$





where $K_{P_P}$, $K_{i_P}$ and $K_{d_P}$ are the proportional, integral and derivative gains, $E_\theta(s)$ is the angular position error ($\theta_{L_{sp}} - \theta_L$ or

$\theta_{R_{sp}} - \theta_R$), $FF_\omega$ and $FF_\alpha$ are the feed-forward gains for the angular speed and acceleration, and $R_\omega(s)$ and $R_\alpha(s)$ are the

angular speed and acceleration setpoints. The electrical current regulator transfer function, on the other hand, is described as

$$C_C(s) = \left( K_{p_C} + \frac{K_{i_C}}{s} \right) E_C(s), \tag{14}$$

where $K_{p_C}$ and $K_{i_C}$ are the proportional and integral gains and $E_C(s)$ is the electrical current error. All parameters have been

tuned using the manufacturer configuration software, and its values are given in Table 3.

As represented by the blocks $K_\ell$ in Fig. 4, the motion of each motor varies the tether distance. To find its transfer function

we divide by $k_{x\theta}$ and use Eq. (2) to find

$$K_{\ell_j}(\theta_j) = \Delta\ell\left( \frac{\theta_j}{k_{x\theta}} \right), \quad j \in \{L, R\}. \tag{15}$$

As $\ell_j(0)$ is identical for both actuators, we directly find $\Delta L_u$ in Eq. (4) by subtracting Eqs. (2). The actual difference in

length $\Delta L$ perceived by the kite is not $\Delta L_u$, but rather the combination of the control input and the so-called geometric control

input. As pointed out by Fagiano et al. (2014), when the output points of the control tethers on the GCU are separated by a

distance $d_c$ (shown in Fig. 1 and value given in Table 1), a difference in length between the tethers is induced for kite's positions

outside the vertical plane spanned by $\boldsymbol{i}_G$ and $\boldsymbol{k}_G$. The geometric control input $\Delta_g$ and the resulting $\Delta L$ are thus modeled as

$$\Delta L_g = d_c \sin\delta \cos\lambda, \quad \Delta L = \Delta L_u - \Delta L_g. \tag{16}$$

## 4   Experimental results

A flight test campaign was carried out on a field near Santa María de la Alameda, located on the Guadarrama mountain range

of Madrid (Spain). Two flight tests have been selected to showcase the effects of the variation of the guidance parameters on the

trajectory and analyze its impact on the dynamics of the RFD kite. The first flight test (denoted as Flight A) was performed with

a moderate wind of 7.2 m/s (standard deviation of 1.5 m/s) and lasted for 321 seconds from the activation of the autonomous

controller. For the second flight test (Flight B), the wind was weaker, with a mean velocity of 5.4 m/s and a standard deviation

of 1.1 m/s, and lasted for 53 seconds. In both flights the kite was piloted manually during the takeoff maneuvers, and the

experiment concluded when the wind speed fell below the operational range of the kite. Tension data was recorded in both

flights by the on-ground tensiometers. The data of the on-board load cells was only available for Flight B due to a sensor

failure during Flight A. Table 2 shows the values of the guidance module's parameters used in both flights.

### 4.1   Performance of the guidance and control modules

Figure 5 shows the trajectory of the kite on the $\delta - \lambda$ plane for Flight A (in gray) and Flight B (in color). Colors magenta, brown,

green and blue were used to represent in Flight B the leftward turns, leftward straight segments, rightward turns and rightward

straight segments, respectively. As shown in the figure, the segments connecting the two turns are not totally straight but, after



**Table 2.** Parameters used for the guidance module state machine for each flight.

| State | Parameter | Value (Flight A) | Value (Flight B) |
|---|---|---|---|
| Straight Right | Attractor point | $\delta_{R_-} = -40°, \lambda_{R_-} = 25°$ | $\delta_{R_-} = -40°, \lambda_{R_-} = 15°$ |
| | Transition condition | $\delta_{L_-} = -25°$ | $\delta_{L_-} = -15°$ |
| Left Up-Turn | Center point | $\delta_{C_-} = -25°, \lambda_{C_-} = 35°$ | $\delta_{C_-} = 0°, \lambda_{C_-} = 25°$ |
| | Transition condition | $\psi_{L_-} = 15°$ | $\psi_{L_-} = 0°$ |
| Straight Left | Attractor point | $\delta_{R_+} = 40°, \lambda_{R_+} = 25°$ | $\delta_{R_+} = 40°, \lambda_{R_+} = 15°$ |
| | Transition condition | $\delta_{L+} = 25°$ | $\delta_{L+} = 15°$ |
| Right Up-Turn | Center point | $\delta_{C_+} = 25°, \lambda_{C_+} = 35°$ | $\delta_{C_+} = 0°, \lambda_{C_+} = 25°$ |
| | Transition condition | $\psi_{L+} = -15°$ | $\psi_{L+} = 0°$ |

the turn, they have two subsegments with opposite convexity. To highlight them, dark and light tonalities were used (brown and blue for the two straight segments). The analysis of the experimental results revealed that the inflection points where the
concavity changes correspond to the condition $d\psi/dt = 0$, i.e. when $\psi$ reaches extreme values. We finally mention that the straight segments and the turns highlighted with colors represent distinct flight conditions which are relevant in the analysis. However, as shown below, they do not necessarily match the states of the finite-state machine of the controller presented in Sec. 3.

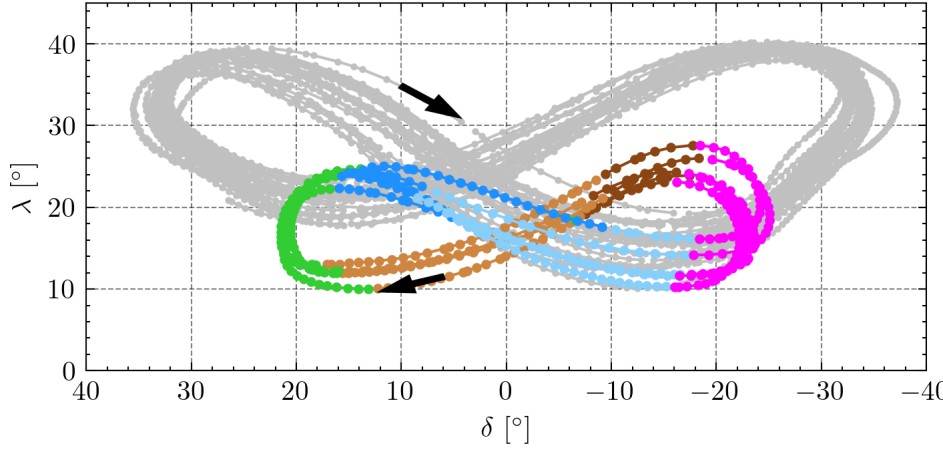

**Figure 5.** Kite trajectory on the $\delta - \lambda$ plane during autonomous operation for Flight A (gray) and Flight B (colored).

Flights A and B are illustrative examples on how by tuning the parameters of the control, an effective adjustment of the
trajectory of the RFD kite can be achieved. During Flight A, the controller was able to perform highly repeatable trajectories, yet overly wide and asymmetric. The right turning maneuver took place too far from the center of the wind window, significantly decreasing tether tension and thus resulting in impaired control capabilities. The lessons learned in Flight A were used to change



| Block | Parameter | Value |
|---|---|---|
| Tether length difference controller | Proportional gain ($K_p$) | 0.23 m/rad |
| | Integral gain ($K_i$) | 0.0 m/(rad · s) |
| | Derivative gain ($K_d$) | 0.006 (m · s)/rad |
| | Decay constant ($d$) | 0.2 |
| | Sampling period ($T$) | 0.1 s |
| Linear to angular converter | Mechanical gain ($k_{x\theta}$) | 1173 rad/m |
| Left motor position controller | Proportional gain ($K_{p_P}$) | 29.52 A/rad |
| | Integral gain ($K_{i_P}$) | 1236.59 A/(rad · s) |
| | Derivative gain ($K_{d_P}$) | 229.30 (mA · s)/rad |
| | Feed-forward velocity gain ($FF_\omega$) | 5.63 (mA · s)/rad |
| | Feed-forward acceleration gain ($FF_\alpha$) | 0.62 (mA · $s^2$)/rad |
| Left motor current regulator | Proportional gain ($K_{p_C}$) | 452.74 mV/A |
| | Integral gain ($K_{i_C}$) | 421.87 V/(A · s) |
| Right motor position controller | Proportional gain ($K_{p_P}$) | 27.60 A/rad |
| | Integral gain ($K_{i_P}$) | 1156.31 A/(rad · s) |
| | Derivative gain ($K_{d_P}$) | 214.06 (mA · s)/rad |
| | Feed-forward velocity gain ($FF_\omega$) | 5.61 (mA · s)/rad |
| | Feed-forward acceleration gain ($FF_\alpha$) | 0.58 (mA · $s^2$)/rad |
| Right motor current regulator | Proportional gain ($K_{p_C}$) | 452.74 mV/A |
| | Integral gain ($K_{i_C}$) | 421.87 V/(A · s) |

**Table 3.** Initial control parameters values. As shown in the table, the tether length difference controller is effectively used as a PD controller for the presented tests.

the configuration of the guidance module in Flight B, which generally exhibited tighter and more symmetric trajectories and sharper turns. Narrower figure-eight paths were also achieved due to lower elevations for all reference points in the guidance
module.

In Flight A, the guidance module was tuned with points $C_\pm$ placed at the same azimuth as the transition points $L_\pm$, i.e we took $\delta_{C\pm} = \delta_{L\pm} = \pm 25°$ (see Table 2) and the elevation of the attractor points $R_\pm$ ($\lambda_{R_\pm} = 25°$) was higher than the elevation of the kite at the straight-to-turn transition points. A consequence of this configuration was that the kite was already moving upwards when the state machine changed from the straight to the turn navigation phase. Since the heading setpoint in the turn
was set perpendicular to the segment $\overline{C_\pm O_K}$ as explained in Sec. 3, it resulted in a steering command opposed to the one that was needed to make the turn during the first instants of the turning phases, thus delaying the turning maneuver.

In Flight B, the parameters of the guidance module were changed to improve the performance. The only exception is the azimuth of the attractor points $R_\pm$ that were kept equal to $\delta_{R_\pm} = \pm 40°$. As shown in Table 2, the elevation angles of points





$C_\pm$ and $R_\pm$ were decreased $10°$ to lower the height of the figure-eight and make the kite flight more perpendicular to the wind
direction. As shown in Fig. 5, such a configuration successfully lowered the trajectory. Secondly, the azimuth angles of the
straight-to-turn transition points $L_\pm$ were decreased from $\delta_{L_\pm} = \pm25°$ to $\delta_{L_\pm} = \pm15°$ to avoid the kite visiting the edges of
the wind window during the turns. Figure 5 makes evident the impact of such a change on the trajectory of Flight B, whose
azimuth was bounded within the range $-25° < \delta < 25°$. The third and last change was targeted to eliminate the unsatisfactory
steering command at the beginning of the straight-to-turn transition. With this aim, the azimuth angle of points $C_\pm$ was set
$\delta = 0$. The result was a stronger steering input in the appropriate direction.

To get a deeper understanding of the performance of the guidance module and its configuration, Fig. 6 displays the evolution
of the kite heading $\psi$ provided by the on-board computer (colored with the same code as Fig. 5) and the heading angle setpoint
$\psi_{sp}$ used by the controller (black line). Such an angle is given by Eqs. (9) and (10) for the straight and turn phases, respectively.
For each flight, two representative cycles are shown. The timesteps when a transition condition is met and thus the controller
switches states according to Table 2 are marked with dashed vertical lines.

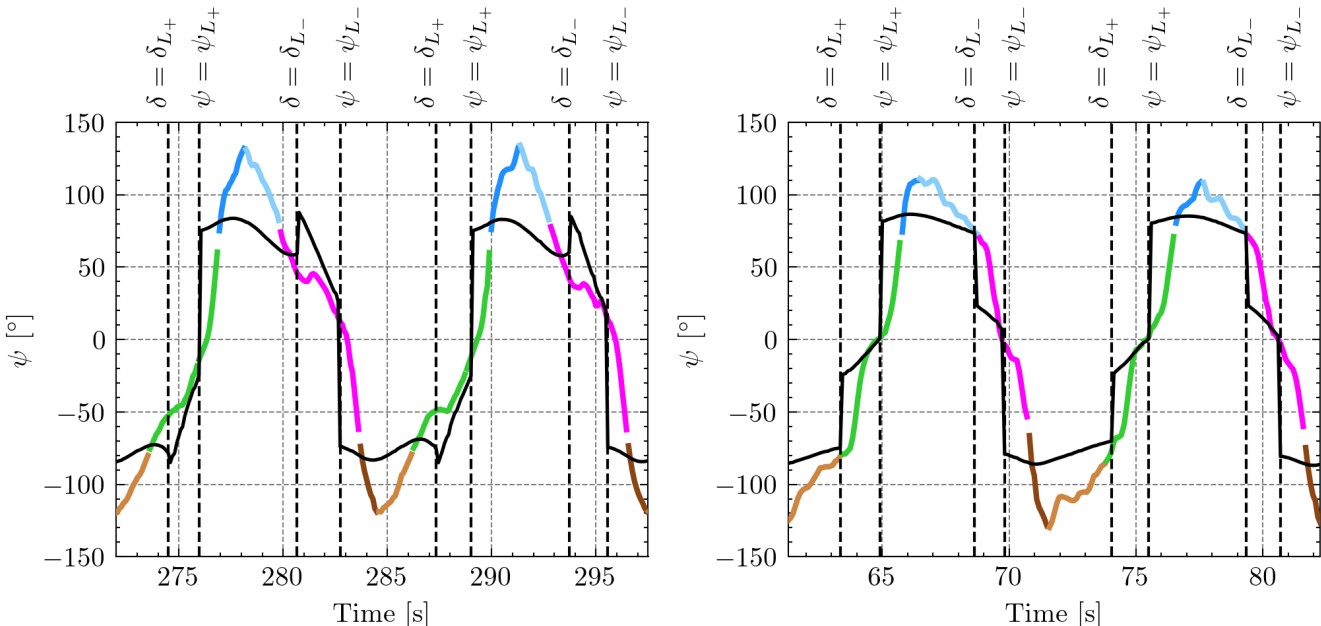

**Figure 6.** Evolution of the heading angle $\psi$ (colored) and the controller setpoint $\psi_{set}$ (black) for Flight A (left) and Flight B (right) for two
figure-eight cycles. Transitions between control states and its conditions are marked with dashed vertical lines. The same color code as Fig.
5 is used.

After analyzing the two flights, three improvements are identified in Flight B when compared to Flight A. Firstly, there is a
significant overshoot during the straight segments for both flights (blue and brown segments in Fig. 6). The maximum value of
the setpoint of the heading angle of the controller is around $\psi_{sp} \approx \pm80°$, but the kite reaches heading angles beyond $\pm110°$.
However, the overshoot is smaller for Flight B. Secondly, it is evident that at the beginning of the straight-to-turn transitions





(from blue to pink) in Flight A, the controller provides a wrong steering command (for instance $\psi_{sp}$ increases at $\delta = \delta_{L_-}$ instead of decreasing). This issue was corrected by the new guidance parameters of Flight B. Notably, almost no overshoot is present during the turning segments. For Flight B, the matching between $\psi$ and $\psi_{sp}$ is excellent during the last part of the turnings. Since there is no overshoot in the turns and a small overshoot for Flight B in the straight segments, these results suggest implementing different control gains for the tether length controller (gains $K_p$, $K_i$ and $K_d$ in Table 3) for each phase

and for each wind speed. The third improvement is related to the timing of the transitions. For Flight B, the straight-to-turn transitions of $\psi$ (from blue to pink and from brown to green segments) are very well synchronized with the transition of the state machine controller given by $\psi_{sp}$. The lack of synchronization, the larger value of $\delta_{L_\pm}$, and the misalignment of the GCU with respect to the direction of the wind, yielded wider figures-of-eight trajectories for Flight A.

Previous works on the control of two-line soft kites (Fagiano et al., 2014) (Wood et al., 2015) and RFD kites (Castro-

Fernández et al., 2023) found the following linear dependence between $\Delta L$ and the derivative of the course angle $\dot{\gamma}$

$$\dot{\gamma}(t) = K_L \Delta L(t - t_d), \tag{17}$$

where $K_L$ is the so-called steering gain, and $t_d$ represents the response delay of the kite. These parameters are computed as the values that provide the minimum error of a least-squares fitting of Eq. (17) to each dataset for the points contained in the central part of the wind window (i.e., $|\delta| < 0.17$ rad according to Wood et al. (2015)). Figure 7 shows the results for both

flights, identifying $K_L = -1.7$ rad/(ms) and $t_d = 0.64$ s for Flight A, and $K_L = -0.9$ rad/(ms) and $t_d = 0.28$ s for Flight B.

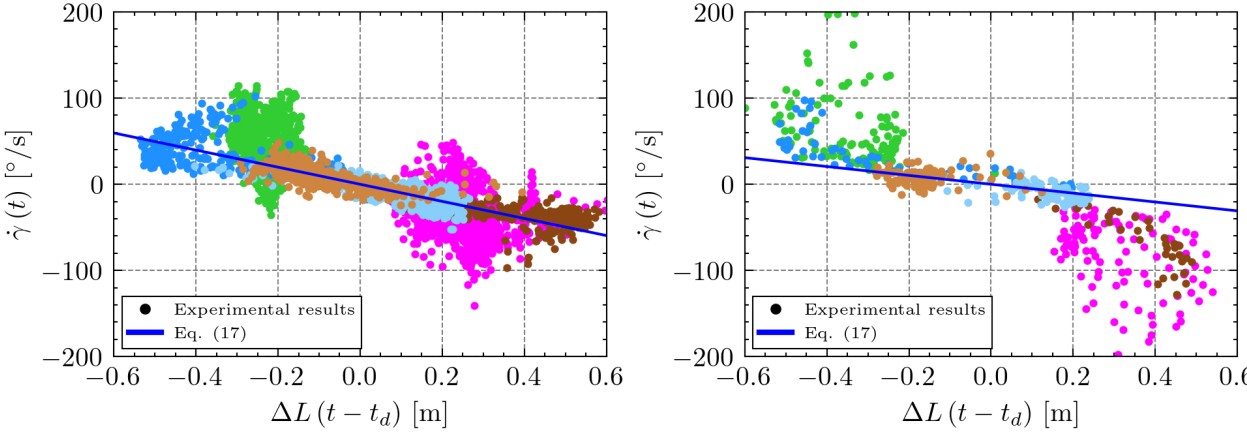

**Figure 7.** Time derivative of the course angle ($\dot{\gamma}$) versus the delayed steering input $\Delta L$ for Flights A (left) and B (right). A least-squares fitting according to Eq. (17) for the points contained in $|\delta| < 0.17$ is represented with a blue line.

The delays obtained by fitting these experimental results have the same order of magnitude than the delay reported by Castro-Fernández et al. (2023) for a similar RFD kite in a two-line configuration that was 0.2 s. In contrast, the steering gains are significantly smaller to $K_L = -8.4$ rad/(ms) found in said work. However, it is worth noting that the criteria of excluding the points outside the central part of the wind window was not used in Ref. Castro-Fernández et al. (2023). The larger drag due





to the third tether in our setup and the different trimming of the central tether needed for three-line operation can explain the slower dynamics. Interestingly, both datasets in Fig. 7 present a significantly smaller dispersion due to the regular nature of the closed-loop trajectories in comparison to the open-loop control of the kite in Castro-Fernández et al. (2023).

## 4.2 Kite and tether dynamics

To get insight into the dynamics of semi-rigid and hybrid kites, such as the RFD kite used for this study, the relationships
between our control variable $\psi$ and different state variables of the kite were analyzed. A strong correlation between $\psi$ and the roll angle ($\phi$) was found, as shown in Fig. 8. In order to model this relationship, we propose the function

$$\phi = A \arctan(-B\,\psi) + C\,\psi, \tag{18}$$

where A, B and C are empirical coefficients related to the amplitude of the roll range, and its rates of change during the turns and the straight segments, respectively. A fitting to the experimental results provides $A = 1.76$, $B = 1.82$ and $C = 0.707$ for
Flight A and $A = 1.53$, $B = 3.85$ and $C = 0.644$ for Flight B. This novel relationship is useful for the future design of three-line RFD kite controllers. An interesting research topic, which is beyond the scope of this work, is its application to two-line RFD kites in order to determine whether the third line has a significant effect on this correlation.

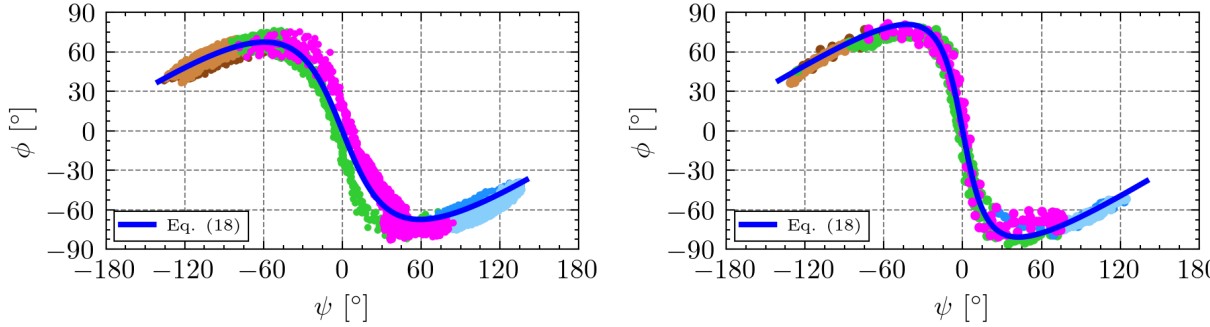

**Figure 8.** Experimental results (dots) and fitting (blue line) of the heading angle ($\psi$) versus the roll angle ($\phi$) for Flight A (left) and Flight B (right).

As shown in Fig. 8, and as advanced in the first paragraph of Sec. 4.1, the extreme values (maximum and minimum) of the heading angle $\psi$ occur within the straight segments and where the dark and light subsegments meet. During the turns (green
and magenta), the steering input induces a change of sign in the roll angle. Flights A and B exhibit some important differences. Firstly, the $\phi - \psi$ curve in Flight A has certain hysteresis within the turns of the path (pink and green color), probably because the turns occur at the extreme of the wind window and the contribution of $\Delta L_g$ to the induced roll is significant. On the contrary, the $\phi - \psi$ curve of Flight B presents a univocal relationship. The slope $d\phi/d\psi$ in the turn, which corresponds to the straight segment in Fig. 8, is higher in Flight B. Since the tether control mainly commands a change in the roll angle for
our three-line RFD kite, we conclude that in Flight B the response of the kite to the command in the turns was weaker. This interesting experimental result can be explained due to the lower wind velocity in Flight B.





A singular characteristic of the experimental setup of this work is the measurement of the on-ground and on-board tether tensions. Figure 9 shows them during three figure-eight cycles of Flight B, with the colored points being the measurements of the on-ground tensiometers and the dashed lines corresponding to the on-board load cells. Panels (a)-(d) correspond to the total

tension, the central, left and right tethers, respectively. The evolution of the tension on the central tether follows closely the behavior of the total tether tension, and it is about one half of its magnitude. This result shows that the control action produces a redistribution of the tensions in the steering (left and right) tethers. The total tension rises during the downwards straight sections, and decrease sharply during the upwards turns, in agreement with the expected behavior of this kind of trajectory (Erhard and Strauch, 2015). The tension on the right (left) control tether increases during the right (left) turns. The maxima in

tension of the left tether corresponds to the minima of the right tether and vice-versa.

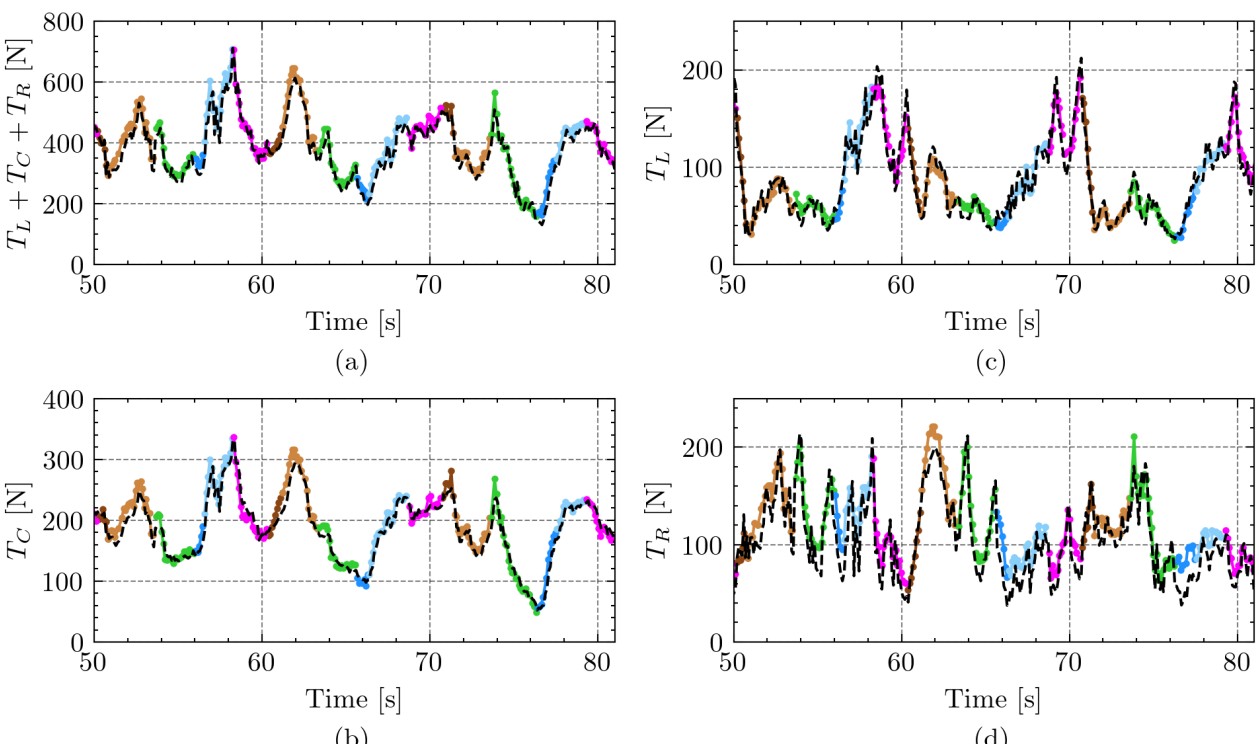

**Figure 9.** Evolution of tether tensions measured by the on-ground tensiometers (colored points with the same code as in Fig. 5) and the on-board load cells (dashed black lines) for three figure-eight cycles of Flight B. Time $t = 0$ is the takeoff.

No significant differences are observed in magnitude between the air and ground load cells. However, a delay ranging between 0.1 and 0.2 s was observed. Such a delay, which only affects the tethers, is smaller than the steering kite gain delay found in Sec. 4.1, which involves the tether and the kite. To understand this tether delay, we estimate the speed of longitudinal





and transversal waves traveling through a tether as

$$V_L = \sqrt{\frac{E}{\rho}}, \quad V_T = \sqrt{\frac{T}{\mu}}, \tag{19}$$

where $E$ is the Young modulus of the tether material, $\rho$ is its volumetric density, $T$ is the mean tension on the tether and $\mu$ is its linear density (French, 1971). These equations yield a longitudinal and transversal wave speed of 13211 m/s and 299 m/s for the central tether during Flight B. As the length of the tethers during flight was approximately 95 m, the characteristic times for a perturbation measured by the on-board load cells to reach the on-ground load cells (or vice versa) are about 7 ms and 0.3 s. As expected, the delay observed in the experimental results is due to the finite velocity of the transversal waves in the tether. Interestingly, the experimental setup is able to capture this important effect and it could be used in future works to study the delay as a function of the tether length and tether sagging for AWE systems.

## 5 Conclusions

The improvements implemented in the GCU and the RFD kite of the small-scale testbed of the UC3M extended considerably the capabilities of the infrastructure and opened new possibilities for its application to the research of AWE systems. These improvements include a modification of the mechanical control system to add a third tether and provide pitch control, the use of running line tensiometers to measure the three tether tensions while allowing for tether reel-in and reel-out, adding on-board load cells attached to the bridle lines to measure the tether tensions on the kite, and a real-time controller for the autonomous flight of the kite. The proposed architecture of the mechanical control system, the electronic system architecture, and the guidance and control modules developed in this work yielded a system capable of performing figure-eight cycles autonomously and consistently while providing valuable scientific data for AWE systems.

A flight campaign with two different flights revealed that the PID controller, which is based on a hybrid guidance strategy that uses attractor points for the straight segments of the cycle and a continuous formulation for the turns, validated the autonomous operation of the testbed for three-line RFD kites. More importantly, it was shown that the shape of the figure-eight cycles, including its lateral amplitude, elevation and radius of the turns, can be chosen by tuning the parameters of the controller. Since the latter are the azimuth and elevation angles of certain characteristic points of the guidance module, and the transition values of the heading angle to change from straight to turn phases, the setting of the parameters of the controller is intuitive. Simple plots of the kite's heading angle and the heading angle setpoint of the controller can be used to tune its parameters. The analysis of the results showed that the performance could be improved even more by applying a gain scheduling control scheme for the straight and the turn phases and wind conditions.

The RFD kite flying autonomously and following highly repeatable figure-eight trajectories allowed to investigate interesting matters related to dynamics of the tethers and the kite. A linear dependence between the control action and the derivative of the course angle was identified and characterized. The analysis reveals a significantly slower dynamic than reported on a previous work for the same RFD kite but in a two-line configuration. The aerodynamic drag on the third line and a different kite trimming can explain it. On the other hand, a strong correlation between the heading and roll angles of the RFD kite was found



and modeled by a simple analytical formula with coefficients found from the experimental data. This correlation may be useful to characterize the dynamic behavior of a kite for a given trajectory and control system. Its application to two-line RFD kites is another open problem to be studied in the future. Finally, the analysis of the tether tensions evolution during the figure-eight trajectories revealed the distribution among the three lines as a function of the actuation. The real-time measurement of the

tether tensions by the on-board load cells and on-ground tensiometers demonstrated that the loads are very similar for our short tether configuration. A variable time delay between 0.1 s and 0.2 s between both measurements associated to the propagation velocity of transversal waves along the tether was identified. Future works can be conducted to correlate this phenomenon to relevant variables in AWE system such as tether sagging and length.

*Author contributions.* F.D.-N, I.C.-F. and G.S.-A. conceptualized the architecture of the mechanical control system. F.D.-N. designed and

built the mechanical control system and developed the software. F.D.-N. and J.G.-G. planned and carried out the experiments. All the authors significantly contributed to the analysis and discussion of the results. F.D.-N. wrote the manuscript in consultation with G.S.-A., J.G.-G and I.C.-F. G.S.-A. supervised the project. All authors have read and agreed to the published version of the manuscript.

*Competing interests.* The authors declare no competing interests.

*Acknowledgements.* This work is part of the project PID2022-141520OB-I00 funded by MICIU/AEI/ 10.13039/501100011033. Work by

F.D.-N. was funded by the grant with reference IND2022/AMB-23521 funded by Comunidad de Madrid.





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
