# Peer review of "A small-scale and autonomous testbed for three-line delta kites applied to airborne wind energy"

_Wind Energy Science, 2024_

## Referee Comment (RC2)

**Review of "A small-scale and autonomous testbed for three-line delta kites applied to airborne wind energy" by Francisco DeLosRíos-Navarrete, Jorge González-García, Iván Castro-Fernández, and Gonzalo Sánchez-Arriaga**

Fernando A. Fontes, University of Porto, Portugal

Submitted to Wind Energy Science, 2024

The article presents an overview of the development of a small-scale, autonomous testbed for an airborne wind energy (AWE) system, composed of a three-line delta kite and a ground station. It provides a thorough description of the mechanical, electronic, and control systems of the testbed. The authors describe the design and construction of the testbed, which is an improvement of a pre-existing system by adding a third tether for pitch control, incorporating tensiometers and on-board load cells for precise tension measurement, and implementing a real-time controller enabling autonomous figure-eight flight trajectories.

The AWE research group at the Universidad Carlos III de Madrid, collaborating with CT Inginieros and INTA, has been working on the development of AWE systems for several years, having provided valuable contributions in theoretical modelling and control results, development of high fidelity simulators, and experimental testbeds, mainly considering systems with soft-wing (possibly with rigid frame) kites. Consequently, the article expresses the authors' mature view and expertise in the field, providing a valuable reference for the AWE community.

The control system is well explained, both regarding the mechanical actuators and the control algorithm. This last is based on a hybrid guidance

strategy, combining attractor points for straight segments and continuous heading angle tracking for turns, which has been shown to be effective in achieving figure-eight paths. The authors present experimental validation of the control scheme, analyzing two flight tests and demonstrating the impact of tuning the guidance parameters on the kite's trajectory. The study includes a detailed analysis of the experimental data, including tether tensions, heading and roll angles, and the relationship between steering input and course angle. The authors also identify and model a correlation between heading rates and roll angles of the kite. The controller relies heavily on the choice of the guidance parameters, which are reported in Table 2. The impact of parameter variations in controller performance could be further investigated. It seems to me that a significant parameter that could be further analyzed is the radius of the circles centred at the attractor points, $C_-$ and $C_+$. This parameter should be larger than the minimum turning radius of the kite (ocurring at the maximum roll angle), and would condition the choice of the other parameters.

Although the development and validation of a small-scale testbed – as well as the data collection it enables – are of great importance for the AWE community, a future commercially viable system would require some modifications that could be worth discussing in this article. These modifications would include not only a larger dimension, but also a reduction on the number of tethers. This is because the number of tethers can significantly hinder the power efficiency of the system. The following example shows that the use of a single tether, instead of three, can be a major advantage, more than doubling the power output of the system.

> Consider a kite system with a single tether and a similar one with three-line tether system. Let the system aerodynamic parameters be as reported in [1], with the well-studied Makani Wing 7, an 8 m wingspan, 30 kW prototype, where the aircraft drag coefficient $C_{Da}$ is 0.06 and the tether drag coefficient $C_{Dt}$ is 0.11 using a 160 m tether length (20 times the wingspan). The total drag coefficient for this situation is $C_{D1}$=0.17.

> Suppose that 3 tethers, instead of one are used. If the total cross section area of the tether is maintained (assuming the same total maximum load), each of the 3 tethers should have a cross section with radius $R/\sqrt{(3)}$ when $R$ was the radius of the section of the previous tether. Since the drag of the tether is proportional

to the radius of its cross section, then the drag of the 3 tethers is $\sqrt{(3)} = 1.73$ times larger, resulting in total drag of $C_{D3} = 0.06 + 1.73 * 0.11 = 0.25$.

A well-known formula for the estimation of the power output $P$ of a kite system establishes that the maximum power extracted is in an inverse proportion to the square of the total drag. Therefore, denoting by $P_{\text{single}}$ and $P_{\text{three}}$ the power output of the single tether and three-line tether systems, respectively, we have

$$\frac{P_{\text{single}}}{P_{\text{three}}} = \left(\frac{C_{D3}}{C_{D1}}\right)^2 = 2.16$$

That is, all the rest being equal, the maximum power that can be extracted using a single tether is more than the double of the power that can be extracted using a three-line tether system.

The authors could consider discussing the possibilities of using a single tether system, possibly having a single tether that splits into a variable geometry bridle, as is used in the Kitepower/ TU Delft system [ref] with a hanging control pod, or as in the University of Porto/Upwind project system [ref] with actuators for varying the bridle geometry inside the aircraft.

The authors may also want to consider discussing the possibility of using other control strategies, that could be more efficient in terms of power output than the one proposed in the article, such as the ones that use optmization-based methods and try to follow a trajectory that maximizes average power generated.

The potential for scaling up the testbed and the predicted challenges associated with such modifications are also a relevant research aspect and a discussion of some of these questions would make the article even more interesting to the AWE community.

**References**

[1] Damon Vander Lind. Analysis and flight test validation of high performance airborne wind turbines. In *Airborne Wind Energy*, pages 473–490. Springer, Berlin, Heidelberg, 2013.

---

## Author Comment (AC1)

*In this paper, the authors presented the system architecture of their ground station AWE testbed, followed by some practical discussions on tuning the controller to track a figure of eight motion. The paper is well-written and provides some interesting insights for further research.*

We are glad to know that the Reviewer found that our work provides interesting insights, and thank them for providing the minor comments we address below.

*MINOR COMMENTS*
*1. The discussions surrounding the individual elements of each flight (lines 251-265) are a bit hard to follow. This makes it difficult to follow subsequent discussions, which are based on lines 251-265. I suggest keeping figure 5 but adding two more annotated figures for flights A and B, then expanding lines 251-265 to describe the components of flight A and flight B trajectories separately.*

In the reviewed version of the manuscript, Figure 5 has been subdivided into two subfigures representing Flights A and B separately, and the attractor and center points have been added to the figure. Additionally, Figure 5 and Tables 2 and 3 have been rearranged to present all relevant information to the discussion next to each other. We thank the Reviewer for this suggestion, which improved the quality of the manuscript.

*2. The authors mentioned the addition of a third tether line for pitch control, but there's not much (if any) discussion on the improved pitch dynamics. I assume that the window to do pitch dynamics flight tests has passed. If so, some comments on how the third line can improve flying qualities and/or direction for future studies regarding this would be appreciated.*

As pointed out by the Reviewer, the pitch control was not incorporated into the autonomous controller but rather tuned by the human operator if necessary. Following the Reviewer's advice, a paragraph has been added at the end of Sec. 3 (line 226) noting this circumstance and discussing future extension of the controller to leverage these capabilities present in the hardware. Our group has updated the Ground Station (we are now in the integration phase), and in the next flight campaign the pitch control will be tested.

*TECHNICAL COMMENTS*
*Some suggested typo/grammar fixes:*
*1. Line 83: we refer to CL and CR as the points where*
*2. Line 134: please format the reference*
*3. Line 291: same order of magnitude as*
*4. Line 308: and as noted in the first paragraph*
*5. Line 321: double use if it/its. Unsure which one is which.*
*6. Line 331: Young's*

We thank the Reviewer for the thorough revision of our manuscript and the useful comments that improved the quality of our work. The typos were all fixed, including some additional ones which were detected in Eq. 10 and Table 2 during the revision of the manuscript.

---

## Author Comment (AC2)

**Referee Comment #2 - WES-2024-170**

We thank Prof. Fontes for the thorough revision of our manuscript and his comments and suggestions.

*The AWE research group at the Universidad Carlos III de Madrid, collaborating with CT Inginieros and INTA, has been working on the development of AWE systems for several years, having provided valuable contributions in theoretical modelling and control results, development of high fidelity simulators, and experimental testbeds, mainly considering systems with soft-wing (possibly with rigid frame) kites. Consequently, the article expresses the authors' mature view and expertise in the field, providing a valuable reference for the AWE community.*

We are glad to know this positive opinion of Prof. Fontes about this work and the contribution of UC3M group to AWE.

*The impact of parameter variations in controller performance could be further investigated. It seems to me that a significant parameter that could be further analyzed is the radius of the circles centred at the attractor points, $C-$ and $C+$. This parameter should be larger than the minimum turning radius of the kite (ocurring at the maximum roll angle), and would condition the choice of the other parameters.*

We fully agree with Prof. Fontes that there is still room for exploring the impact of the parameters of the controller into the kite trajectory. It is a very interesting research problem, but beyond the scope of this work that it is focussed on presenting a new ground station, a controller, and experimental results for two complete sets of parameters (see Table 3). The radius of the circle is not a parameter that we could directly impose in the controller because the guidance module is based on the kite attitude (and not on its position). Nonetheless, this suggestion encourages us to investigate in future flight campaigns the dependence between the radius of the turn with the parameters of the controller that we can directly define like the position of points C, L and R. In particular, we could try to find the flight envelope (minimum turning radius) as a function of the wind velocity by progressively changing the controller parameters to reduce the radius. Such a research activity is fully aligned with the purpose of the testbed presented in this manuscript, that is using the infrastructure to investigate basic dynamic and control phenome in AWE systems.

This important point has been now mentioned in the new paragraph at the end Sec. 4.1 (line 305).

*Although the development and validation of a small-scale testbed – as well as the data collection it enables – are of great importance for the AWE community, a future commercially viable system would require some modifications that could be worth discussing in this article. These modifications would include not only a larger dimension, but also a reduction on the number of tethers. This is because the number of tethers can significantly hinder the power efficiency of the system.*

We fully agree with this comment and also with the later discussion made by Prof. Fontes about 1-line and 3-line AWE systems. For this reason the UC3M testbed has been designed to be compatible with 3-line and 1-line AWE systems. The former is presented in this manuscript and, regarding the latter, we are currently working on a fly-actuated system to work with 1-line AWE systems. Our main objective when developing our testbed (and also its

scaleup version, see below) was not power production but having a flexible platform in terms of system architecture and configuration to conduct research on AWE energy.

*The authors could consider discussing the possibilities of using a single tether system, possibly having a single tether that splits into a variable geometry bridle, as is used in the Kitepower/ TU Delft system [ref] with a  hanging control pod, or as in the University of Porto/Upwind project system [ref] with actuators for varying the bridle geometry inside the aircraft.*

We fully agree with the comment. UC3M AWE research group is currently working on an on-board control system, with a single tether connecting the kite with the ground station due to the reasons pointed out by Prof. Fontes (see Ref. [01]). Such a control system is not based on a pod, but on a mechanical system integrated in the kite that varies the bridle geometry. We recently finished its integration phase and we plan to start a testing campaign next month. As explained in this manuscript, our goal is that the testbed will be a useful platform to explore different types of AWE systems (single and multi tethered machines and flexible and rigid wings).

*The authors may also want to consider discussing the possibility of using other control strategies, that could be more efficient in terms of power output than the one proposed in the article, such as the ones that use optimization based methods and try to follow a trajectory that maximizes average power generated.*

We agree that the original manuscript was not linked enough to the final goal of any AWE machine, which is power generation. Following this suggestion, we added a paragraph at the end of Sec. 4.1 (line 305) discussing the possibility of extending the proposed control strategy for the optimization of power output, and we added a few more sentences to the paragraph starting in line 40 of the Introduction discussing the limitations and advantages of this approach. We thank the Reviewer for this suggestion, which improved the quality of the manuscript.

*The potential for scaling up the testbed and the predicted challenges  associated with such modifications are also a relevant research aspect and  a discussion of some of these questions would make the article even more  interesting to the AWE community*

We fully agree and our research group has already updated and scale-up the testbed presented in this manuscript. We added several new capabilities like for instead mechanical-to-electric power conversion, and upscaled the actuators to work with larger kites (see [02]). Nonetheless, the architecture and methods of the scaled-up testbed are very close to the ones presented in this manuscript. We have recently finished the integration phase and will perform test campaigns in the next months.

We added some sentences at the end of the Conclusions (line 386) to keep the AWE community informed about our roadmap and level of development.

[01] González-García, J., et al. An Aircraft-Integrated Control System Based on Bridle Actuation for AWE Machines. Airborne Wind Energy Conference 2024 (AWEC 2024). DOI: 10.13140/RG.2.2.19533.14569
[02] DeLosRíos-Navarrete, F., et al. A Small-Scale and Multipurpose Airborne Wind Energy Prototype. Airborne Wind Energy Conference 2024 (AWEC 2024). DOI: 10.13140/RG.2.2.15230.70729

---

## Author Comment (AC3)

**Referee Comment #3 - WES-2024-170**

*Summary*
*The paper presents the significant extension of an existing small-scale test bed for airborne wind energy.*

*The choice of wing structure is a rigid-framed delta kite, a concept which is also being pursued by the company Enerkite, one of the more advanced AWE companies. The kite is controlled from a ground station via three lines, allowing steering and pitching of the wing and reeling in and out. Next to the advancements with respect to hardware (adding the third line, line tensionmeters and load cells), the main contribution is the development and experimental validation of a PID-controller based on a mixed waypoint/continuous reference path. The flight experiment data are then utilized to develop models for some of the system subcomponents.*

*The paper is written very well and provides an extensive overview of the existing literature and state-of-the-art. Overall, it is a novel and interesting addition to the existing body of experimental AWE work and can provide a useful resource for practitioners and researchers alike.*

We are glad to know this positive opinion of the Reviewer about our work.

*Minor comments:*
*- Some more context could be provided to motivate the choice of control strategy. Why was this specific strategy chosen, what are its advantages and limitations, in particular in comparison with existing (published) strategies for delta kites?*

We thank the Reviewer for raising this point because probably it was not highlighted enough in the Introduction. To the best of the authors' knowledge, most of the previous scientific articles on AWE guidance and control was focussed on single-line soft kites and fixed-wing aircraft and the available (open) literature on multi-tethered delta kites is scarce. We decided to choose a guidance strategy based on attractor points because it is simple and it has been extensively validated in several experimental setups using leading edge inflatable kites (both ground-actuated and fly-actuated). The main disadvantage is that the kite does not follow a prescribed trajectory (although one could adjust the actual trajectory to a prescribed one by tuning the parameters of the guidance module).

Following this comment by the Reviewer, we added a few more sentences to the paragraph starting in line 40 of the Introduction.

*- While the third kite line for pitch control (as well as the option for reeling in and out) is mentioned as a contribution of the paper, I could not find a corresponding block in the control block diagram (Fig. 4). It would therefore be useful to devote a paragraph outlining a possible extension of the controller to allow for these extra built-in features of the hardware setup.*

As pointed out by the Reviewer, the pitch control was not incorporated into the autonomous controller but rather tuned by the human operator if necessary. Following the Reviewer's advice, a paragraph has been added at the end of Sec. 3 (line 226) noting this circumstance and discussing future extension of the controller to leverage these capabilities present in the hardware. Our group has updated the Ground Station (we are now in the integration phase), and in the next flight campaign the pitch control will be tested.